# Water Column Microbial Communities Vary along Salinity Gradients in the Florida Coastal Everglades Wetlands

**DOI:** 10.3390/microorganisms10020215

**Published:** 2022-01-20

**Authors:** Peeter Laas, Kelly Ugarelli, Rafael Travieso, Sandro Stumpf, Evelyn E. Gaiser, John S. Kominoski, Ulrich Stingl

**Affiliations:** 1Fort Lauderdale Research and Education Center, Department of Microbiology & Cell Science, Institute of Food and Agricultural Sciences, University of Florida, Davie, FL 32611, USA; peeter.laas@taltech.ee (P.L.); k.ugarelli@ufl.edu (K.U.); 2Department of Marine Systems, School of Science, Tallinn University of Technology, 12618 Tallinn, Estonia; 3Institute of Technology, Faculty of Science and Technology, University of Tartu, 50411 Tartu, Estonia; 4Institute of Environment, Florida International University, Miami, FL 33199, USA; travieso@fiu.edu (R.T.); sstumpf@fiu.edu (S.S.); gaisere@fiu.edu (E.E.G.); jkominos@fiu.edu (J.S.K.); 5Department of Biological Sciences, Florida International University, Miami, FL 33199, USA

**Keywords:** Florida Everglades, coastal microbiology, bacterioplankton, eukaryotic microbial communities, 16S rRNA amplicon sequencing, 18S rRNA amplicon sequencing

## Abstract

Planktonic microbial communities mediate many vital biogeochemical processes in wetland ecosystems, yet compared to other aquatic ecosystems, like oceans, lakes, rivers or estuaries, they remain relatively underexplored. Our study site, the Florida Everglades (USA)—a vast iconic wetland consisting of a slow-moving system of shallow rivers connecting freshwater marshes with coastal mangrove forests and seagrass meadows—is a highly threatened model ecosystem for studying salinity and nutrient gradients, as well as the effects of sea level rise and saltwater intrusion. This study provides the first high-resolution phylogenetic profiles of planktonic bacterial and eukaryotic microbial communities (using 16S and 18S rRNA gene amplicons) together with nutrient concentrations and environmental parameters at 14 sites along two transects covering two distinctly different drainages: the peat-based Shark River Slough (SRS) and marl-based Taylor Slough/Panhandle (TS/Ph). Both bacterial as well as eukaryotic community structures varied significantly along the salinity gradient. Although freshwater communities were relatively similar in both transects, bacterioplankton community composition at the ecotone (where freshwater and marine water mix) differed significantly. The most abundant taxa in the freshwater marshes include heterotrophic *Polynucleobacter* sp. and potentially phagotrophic cryptomonads of the genus *Chilomonas*, both of which could be key players in the transfer of detritus-based biomass to higher trophic levels.

## 1. Introduction

Coastal environments, which are among the most diverse and productive habitats in the world, provide ecosystem services worth trillions of dollars annually [1,2]. Over the last century, freshwater diversion and the conversion of wetlands to agricultural land has reduced the area of global coastal ecosystems by more than two-fold [3]. These developments have made the ecological well-being of coastal ecosystems a pressing global issue, and especially conservation of unique and iconic sites that are inscribed on the UNESCO World Heritage List and considered in ‘critical’ status, like the Everglades National Park (ENP), are in the focus of local and national environmental policies. Consequently, the Comprehensive Everglades Restoration Plan, with an approximate 50-year construction schedule, was initiated in the 1990s to restore the quantity, timing and distribution of the pre-drainage water flow through the Everglades that was changed profoundly due to urban development, agriculture and extensive drainage [4,5].

Coastal environments are heavily affected by a heterogeneous variety of terrestrial, freshwater and marine aquatic ecosystems and therefore represent highly complex and dynamic habitats. This complexity challenges informed management decision-making that relies on sufficient knowledge on the state of the system and its responses to external environmental pressures. In the Everglades specifically, shifts in fresh and marine water supply influence total phosphorous concentrations, salinity, and inundation, which in turn affect the net carbon balance in the ecosystem along the freshwater-marine gradient [6]. Most of the current ENP landscape is less than 1.5 m above the mean sea level [7], which makes it highly susceptible to a projected sea level rise of up to 2 m by 2100 [8,9]. Flower et al. [10] estimated that a 0.5 m rise in sea level would lead to seawater and mangroves encroaching 15 km into ENP by 2060. In comparison, paleoecological records demonstrate that in the past 3000 years, the gradual replacement of freshwater marsh by mangrove forests has shifted the ecotone only 20 km inland [11].

While there is considerable effort in using algae as bioindicators for the ‘health’ of aquatic ecosystems worldwide [12], and in the Everglades [13,14,15], prokaryotic communities have largely been overlooked by environmental monitoring programs [16], despite being essential for the functioning of all aquatic ecosystems and food webs [17,18]. Prokaryotic communities respond rapidly to environmental changes and therefore provide valuable bioindicators, which can predict short- to medium term regime shifts of an entire ecosystem [19,20]. Especially the use of high-throughput sequencing has revolutionized studies on microbial communities and has made effective monitoring of microbial communities feasible [19]. In the Everglades, to date only sediment bacterial communities have been analyzed using low-throughput molecular methods. These studies indicated differences in community composition between ecosystem types along the salinity gradient from freshwater to marine habitats. The salinity and total phosphorous in the water column were identified as the main factors explaining these shifts [21].

There are two main major water drainages in ENP: Shark River Slough (SRS), a large long-hydroperiod and peat-based drainage that receives freshwater from the Tamiami Canal and flows directly into the Gulf of Mexico (GoM; Figure 1), and Taylor Slough Panhandle (TS/Ph), a smaller short-hydroperiod and marl-based drainage, located in the eastern boundary of the ENP that is indirectly connected to the GoM through vast seagrass meadows in Florida Bay (FB). Here, we use high-throughput amplicon sequencing of 16S and 18S rRNA genes to determine the composition of bacterial and eukaryotic microbial communities in surface waters along the SRS and TS/Ph transects. Our main questions were: (1) who are the key members of planktonic microbial communities in the Everglades, and (2) how do environmental variables shape these communities along the transects from freshwater to marine habitats? This analysis of microbial communities in the Everglades provides crucial information about an essential part of its food web and serves as a reference to study the impact of environmental change in this unique ecosystem.

## 2. Materials and Methods

### 2.1. Sample Collection

Two liters of surface water samples were obtained from 14 FCE LTER core sampling sites during the subtropical wet season in August of 2017 (Table 1; Figure 1; https://fce-lter.fiu.edu/research/sites/, accessed on 14 December 2021). Samples were kept on ice until filtration and were processed within 12 h. To determine bacterial abundances (BA), 9 mL subsamples were separated for flow cytometry and fixed with paraformaldehyde (final concentration 1%; pH = 7.4), incubated at room temperature (RT) for 60 min, and stored at −20 °C until analyses. The microbial communities were separated during filtration into two size fractions using 5 µm (>5 µm fraction) and 0.22 µm (0.22 µm–5 µm fraction) nitrocellulose membranes (MF-Millipore, Darmstadt, Germany). The physicochemical background data and bacterial secondary production values were obtained via standard FCE-LTER protocols (https://fcelter.fiu.edu/data/protocols/index.html, accessed on 14 December 2021). Stations 1–3 in SRS and stations TS/Ph 1–3 are in freshwater marshes. The remaining three stations in SRS (4–6) and stations 6 and 7 of TS/Ph are considered ecotones. Stations TS/Ph 9–11 are situated in Florida Bay. Details on the sites, including coordinates, can be found at: https://fce-lter.fiu.edu/research/sites/index.php (accessed on 114 December 2021).

### 2.2. Flow Cytometry

Samples for flow cytometry were incubated with SYBR Green I nucleic acid stain for 30 min at RT. Flow cytometry analyses were performed on a Guava easyCyte HT (Luminex, Austin, TX, USA) at a flow rate of 0.24 μL s^−1^. Cell populations were discriminated via green fluorescence (532 nm) and side scatter channels using a blue laser (488 nm). High nucleic acid and low nucleic acid content bacterial cell counts were pooled together to obtain total bacterial abundances (BA, [22]). Samples were analyzed using Guava’s InCyte software (Luminex, Austin, TX, USA).

### 2.3. Molecular Methods

DNA extractions were carried out with the Qiagen PowerWater Kit following the manufacturer’s recommended protocol (Qiagen, Hilden, Germany). DNA sequence data was generated using Illumina paired-end sequencing (151 bp × 12 bp × 151 bp MiSeq run) at the Environmental Sample Preparation and Sequencing Facility at Argonne National Laboratory (Lemont, IL, USA). DNA extracts were used as templates for the amplification of the V4 hypervariable region of the 16S rRNA gene (515F-806R primer pair, [23]) and V9 hypervariable region of the 18S rRNA gene (1389F-EukB primer pair, [24]). In addition, primers contained sequencer adapter sequences and the reverse amplification primer also contained a twelve base barcode sequence for multiplexing. Each 25-µL PCR reaction contained 9.5 µL of MO BIO PCR Water (Certified DNA-Free), 12.5 µL of QuantaBio’s AccuStart II PCR ToughMix (2× concentration, 1× final), 1 µL Forward Primer (5 µM concentration, 200 pM final), 1 µL Golay barcode tagged Reverse Primer (5 µM concentration, 200 pM final), and 1 µL of template DNA. The PCR conditions to amplify the 16S rRNA gene were as follows: 94 °C for 3 min to denature the DNA, with 35 cycles at 94 °C for 45 s, 50 °C for 60 s, and 72 °C for 90 s; final extension of 10 min at 72 °C to ensure complete amplification. The PCR conditions to amplify the 18S rRNA gene were as follows: 94 °C for 3 min to denature the DNA, with 35 cycles at 94 °C for 45 s, 57 °C for 60 s, and 72 °C for 90 s; final extension of 10 min at 72 °C. Amplicons were then quantified using PicoGreen (Invitrogen, Waltham, MA, USA) and a plate reader (Infinite 200 PRO, Tecan). Once quantified, volumes of each of the products were pooled into a single tube so that each amplicon is represented in equimolar amounts. This pool was then cleaned up using AMPure XP Beads (Beckman Coulter, Brea, CA, USA), and afterwards quantified using a fluorometer (Qubit, Invitrogen). After quantification, the molarity of the pool was determined and diluted to 2 nM, denatured, and diluted to a final concentration of 6.75 pM with a 10% PhiX spike for sequencing on an Illumina MiSeq.

### 2.4. Bioinformatics

The QIIME 2 microbiome analysis package [25], was used for sequence analyses. Quality filtering, chimera identification and merging of paired-end reads was carried out using the DADA2 plugin [26], as implemented in QIIME2. SILVA release 132 (Ref NR 99) taxonomy and q2-feature-classifier were used for classification of 16S rRNA gene sequences [27,28]. Data filtering and statistical analyses were carried out with R version 3.2.0 (R Core Team 2014). Vegan package was used to carry out permutational multivariate analysis of variance using distance matrices (vegdist and adonis functions, with 999 permutations) and perform detrended correspondence analyses (decorana function) with environmental fitting (envfit function) [29]. Sequence variants (SVs) classified as chloroplasts or mitochondria were discarded from the dataset. Demultiplexed raw data was submitted to the Sequence Read Archive under accession number PRJNA525456.

### 2.5. Preprocessing of the Dataset

A total of 15,698 sequence variants (SVs) of microbial eukaryotes and bacteria were obtained from the analyses of 14 water samples (Table 2). Amplicon sequencing of the 16S rRNA gene (V4 hypervariable region) was used to analyze bacterioplankton community composition (BCC) within the 0.22–5.0 µm size fraction. After quality control and the removal of mitochondrial and chloroplast ribosomal rRNA sequences, a total of 1,207,635 partial 16S rRNA gene sequences were utilized in this study (average of 83,890 sequences per sample, Table 2). All samples were rarefied to 50,000 sequences for statistical analyses. A total of 4755 prokaryotic sequence variants (SVs) were observed, which could be assigned to 562 genus-level taxa (prokaryotic genera, PG, as defined by classification level 6 in QIIME2), out of which 75 contributed more than 0.1% of the sequences in the entire dataset.

The composition of eukaryotic communities was determined via 18S rRNA gene amplicon sequencing (>5 µm size fraction). After discarding all sequences classified as Bacteria and Metazoa, a total of 3,201,328 sequences (average 218,606 per sample, Table 2) was used for further analyses. A relatively large fraction of eukaryotic SVs (10,943) remained classified only as Eukaryota (11.0%). To improve these classifications, eukaryotic sequence variants were clustered into operational taxonomic units (OTUs, with 95% sequence similarity threshold), which lowered the fraction of unclassified eukaryotes to 5.4%. There were 55 OTUs (out of a total of 5020) that contributed to at least 1% of sequences in at least one of the samples.

## 3. Results

### 3.1. General Overview of the Datasets

In both transects, bacterial and microbial eukaryotic communities were clearly different among freshwater, ecotone, and marine wetlands. Freshwater communities of both transects were similar to each other, while microbial communities in the more saline samples (ecotone, mangroves), and in Florida Bay, were distinctly different (Figure 2). The bacterial community composition (BCC) was most significantly correlated with salinity (R^2^ = 0.92; *p* < 0.001), concentrations of total nitrogen (TN; R^2^ = 0.66; *p* < 0.01), and chlorophyll *a* (Chl *a,* R^2^ = 0.63; *p* < 0.01; Figure 2). Permutational multivariate analysis using distance matrices was used to quantify the effects of environmental factors, transects (SRS, TS/Ph and FB) and ecosystem type (freshwater marshes, ecotone and marine). The ecosystem type better explained variability in BCC (R^2^ = 0.44; *p* < 0.001) than salinity (R^2^ = 0.10; *p* < 0.05), which was the only environmental variable with significant effect (Appendix A). When the ecosystem types in different transects were split into separate groups, even more variability in BCC was explained (R^2^ = 0.58; *p* < 0.001). 

At a higher taxonomic level, BCC followed patterns documented for other estuarine and wetland ecosystems with a strong salinity gradient [30,31], with *Betaproteobacteria* and *Actinobacteria* dominating low salinity environments, and *Alpha*- and *Gammaproteobacteria* dominating coastal marine environments. Relative abundances of the *Bacteroidetes* phylum stayed relatively constant in both fresh and saltwater environments (Figure 3). Although the dataset contained 101 different class-level bacterial taxa, 94.4% of sequences could be assigned to just ten higher taxa: *Betaproteobacteria* (39.2%), *Alphaproteobacteria* (14.4%), *Actinobacteria* (13.6%), *Gammaproteobacteria* (10.3%), *Flavobacteriia* (5.7%), *Sphingobacteriia* (3.2%), *Spartobacteria* (3.2%), *Cyanobacteria* (2.3%), *Proteobacteria Incertae Sedis* (1.9%) and *Acidimicrobiia* (0.6%) (Figure 3). In total, 562 genus-level taxa were identified, out of which 75 contributed to more than 0.1% of all the sequences and can therefore be considered ‘common’ in the ecosystem (Figure 4).

Like the BCC, the composition of the eukaryotic microbial communities (EMC) also varied along the physicochemical gradients (Figure 2). Permutational multivariate analysis demonstrated that in the case of the EMC the ecosystem type (R^2^ = 0.27; *p* < 0.001; Appendix A) also outperformed salinity (R^2^ = 0.11; *p* < 0.05) and Chl *a* (R^2^ = 0.09; *p* < 0.05) which there were only two environmental variables with a significant effect. Differences between the transects were not significant. However, discriminating ecosystem types between transects constrained more variability than ecosystem types alone (R^2^ = 0.41; *p* < 0.005).

Most sequences that could be assigned to eukaryotic taxa included members of *Ochrophyta* (27.3%), *Cryptophyta* (15.0%), *Ciliophora* (14.3%), *Dinoflagellata* (13.2%), *Chlorophyta* (2.3%) and *Fungi* (2.3%) (Figure 3). Clear shifts in community composition were evident even for higher taxonomic levels in both BCC and EMC. Therefore freshwater, ecotone and marine estuary ecosystems are discussed separately below for a more comprehensive overview.

### 3.2. Freshwater Marsh Communities

Stations 1–3 in SRS and stations TS/Ph-1–2 are in freshwater marshes. TS/Ph-3 is heavily impacted by water flow (tides and freshwater input), had a very low salinity at the time of sampling (Table 1), and was thus clustered with the other freshwater sites. The lowest bacterial cell abundances were found in freshwater stations compared to the other stations, in both transects (in TS/Ph-1–3 around 1.2 × 10^6^ cells L^−1^ and 1.0 × 10^6^ up to 1.4 × 10^6^ cells L^−1^ in SRS 1d–3; Figure 1). Bacterial secondary production (BP), a proxy for the integration of DOM into bacterial biomass, did not exhibit a clear trend and varied between the freshwater stations (Figure 1). In TS/Ph, BP rates decreased about five times from TS/Ph-1 (36.5 μg-C L^−1^ d^−1^) to TS/Ph-3, while the highest values for the entire dataset were observed in SRS3 (95.8 μg-C L^−1^ d^−1^) (Figure 1).

Most of the 16S rRNA gene sequences in the freshwater samples were assigned to the orders *Betaproteobacteriales* (63.4%), *Chthoniobacterales* (*Verrucomicrobiae*, 7.3%), *Frankiales* (*Actinobacteria*, 7.0%), *Rhizobiales* (*Alphaproteobacteria*, 4.4%) and *Micrococcales* (*Actinobacteria*, 3.9%) (Figure 3). At the genus level (Qiime2 classifier, level 6), core taxa (mean relative abundance over 1% in all marsh sites) were *Polynucleobacter* (26.1%), MWH-UniP1 aquatic group (21.7%, *Burkholderiales*), FukuN18 freshwater group (7.0%, *Verrucomicrobia*), hgcI clade (5.0%, *Sporichthyaceae*), *Limnobacter* (4.0%, *Burkholderiaceae*), *Candidatus* Aquilina (2.0%, *Microbacteriaceae*), *Alsobacter* (2.0%, *Rhizobiales*), NS11-12 marine group (1.9%, *Sphingobacteriales*), *Sediminibacterium* (1.4%, *Chitinophagaceae*), *Candidatus* Planktoluna (1.7%, *Microbacteriaceae*), unclassified *Methylophilaceae* (1.1%), and *Candidatus* Methylopumilus (1.1%, *Methylophilaceae*) (Figure 4).

The freshwater EMC were dominated by *Chryptophyta*/*Cryptomonads* (on average 29.3%), followed by *Ochrophyta* (21.5%), *Ciliophora* (12.4%), *Dinoflagellata* (6.3%), *Fungi* (4.6%) and *Chlorophyta* (1.2%) (Figure 3). In both transects, cryptomonads dominated the P-limited freshwater areas, yet their relative abundance decreased towards the mangrove ecotone, and they were nearly absent in marine samples. The most abundant OTUs were assigned to *Chilomonas* (OTU1, average 26.2%), and *Cryptomonas* (OTU8, 2.7% and OTU14, 2.4%) (Figure 5).

### 3.3. Ecotone

Stations 4–5 in SRS are situated within mangrove forests and TS/Ph-6 and TS/Ph-7 are located in an estuarine ecotone with mangrove islands [32]. The highest bacterial abundances in our datasets were found in ecotone stations bordering marine estuaries, reaching 3.9 × 10^6^ cells L^−1^ in TS/Ph-7 (Table 2). In the TS/Ph transect, BP peaked in TS/Ph-6 (71.9 μg C L^−1^ d^−1^) and decreased nearly 20-fold towards the estuary. A similar trend was observed in SRS, where rates dropped from 42.9 to 22.6 μg C L^−1^ d^−1^ within the ecotone (Figure 1). 

The major differences in the BCC of the ecotones compared to the freshwater stations were decreases of the relative abundances of *Betaproteobacteria* (to 33.4%), *Sphingobacteriia* (to 3.6%) and *Spartobacteria* (to 0.1%), and concomitant increase in the relative abundances of *Actinobacteria* (to 20.6%), *Gammaproteobacteria* (to 13.6%), *Alphaproteobacteria* (to 7.7%), Proteobacteria Incertae Sedis (to 5.2%) and *Cyanobacteria* (to 3.6%). In parallel, increases in the relative abundances of eukaryotic taxa were found among *Ochrophyta* (to 34.6%) and *Ciliophora* (to 21.7%). Significant differences were also detected among both the BMC and EMC of the two distinct ecotones. Members of *Micrococcales* (*Actinobacteria*) became the second most abundant prokaryotic group in the TS/Ph ecotone (30.4%), while contributing only to 4.9% in waters of the SRS ecotone. Similarly, *Synechococcales* and *Rhizobiales* were found to be more abundant in the ecotone of Taylor Slough (on average 7.7% and 7.6%, respectively), while they both contributed to less than 1% in the Shark River Slough ecotone. A large fraction of BCC at the SRS ecotone were identified as *Ectothiorhodospirales* (17.4%), *Thiotrichales* (8.6%) and *Frankiales* (7.9%), while these three orders were absent from the TS/Ph- ecotone. The relative abundance of *Flavobacterales* were similar between the ecotones, contributing 5.1% at TS/Ph and 7.4% at SRS.

Within EMCs, the relative abundance of *Ochrophyta* increased to over 40% at the SRS ecotone (59.8% in SRS5) and more gradually from TS/Ph-6 to TS/Ph-7 (13.0–27.4%). The fraction of ciliates (*Ciliophora*) also increased in the ecotone to an average of 17.3% at SRS and 32.4% at TS/Ph. The average relative abundance of cryptomonads decreased within ecotones of both transects (Figure 3). Compared to TS/Ph, EMC at SRS contained a larger fraction of diatoms, mainly OTU3 (*Urosolenia*; 21.3%), OTU17 (*Thalassiosira*; 4.5%) and OTU16 (*Pleurosigma*; 3.5%). Heterotrophic protists were more prominent in TS/Ph, especially at TS/Ph-7 (OTU7, *Pelagostrobilidium*, 14.0%; OTU10, *Strombidium*, 11.9%; and raphidophyte OTU18, *Viridilobus*, 12.9%).

### 3.4. Florida Bay

In FB, the BCC was dominated by *Flavobacteriales* (13.8%), SAR11 clade (13.4%), *Rhizobiales* (12.3%), *Oceanospirillales* (10.7%), *Rhodospirillales* (8.1%), SAR86 clade (6.8%), *Micrococcales* (6.4%), *Rhodobacterales* (6.1%), *Puniceispirillales* (4.9%) and *Synechococcales* (4.0%). While organic carbon concentrations were very low in the FB sites, the highest concentration of TOC was found at the central part of the bay, TS/Ph-10 (Table 1), accompanied by the highest bacterial cell counts and BP rates within FB (2.8 × 10^6^ cells L^−1^; 13.8 μg C L^−1^ d^−1^), surpassing the corresponding values of TS/Ph-9 (1.5 × 10^6^ cells L^−1^; 4.0 μg C L^−1^ d^−1^). *Ochrophyta* (average 28.5%, mostly diatoms), *Dinoflagellata* (16.2%), *Ciliophora* (8.8%) and unclassified *Opisthokonta* (3.5%) were the most abundant eukaryotic microbial taxa in these waters. The northeastern part of the bay (TS/Ph-9) had higher inorganic nutrient concentrations (Table 1) and autotrophic diatoms dominated the EMC; *Cyclotella* (OTU4, 27.4%) and *Chaetoceros* (OTU11, 17.8%) were the most abundant genus-level taxa.

The BCC at this location was dominated by *Cohaesibacter* (32.6%) that presumably can decompose casein, cellulose, xanthine and hypoxanthine [33]. In addition, other coastal heterotrophs, like *Microbacteriaceae* DS001 (11.9%) and *Litoricola* (9.5%) and unicellular picocyanobacteria of the genus *Synechococcus* (10.1%), which have been reported to form blooms in FB [34], were abundant.

The accompanying EMC were dominated by dinoflagellates (Figure 3): *Fragilidium* (OTU9, 22.3%) *Pyrophacus* (OTU12, 15.1%), *Prorocentrum* (OTU2, 14.2%) and *Pelagodinium* (OTU20, 9.1%). Members of *Fragilidium* are mixotrophic dinoflagellates and also known to prey upon red-tide and toxic dinoflagellates, including *Prorocentrum* species [35].

## 4. Discussion

### 4.1. Freshwater Marsh Communities

Long-term monitoring data indicated that abrupt and sustained increases in TP and DOC from marine storm surges and severe low-temperature events increase bacterioplankton productivity for extended periods, and that these responses are more pronounced in SRS than in the TS/Ph transect [36]. Despite these striking differences in BP, the composition of the freshwater microbial communities was not significantly different between the two transects.

The abundance of *Betaproteobacteria* in numerous freshwater habitats is heavily affected by salinity [37,38,39,40]. A similar trend is apparent in this study as well. The most abundant *Betaproteobacteria* were composed of two genus-level taxa: *Polynucleobacter* (PG002, 26.0%) and MWH-UniP1 aquatic group (PG001, on average 21.6%). Both taxa belong to the order *Burkholderiales*, together with less abundant PG016 (unclassified) and PG050 (*Limnobacter*), which made up 3.9% and 0.4% of all bacterial sequences in the freshwater marshes, respectively.

*Polynucleobacter*-related sequences were divided between two free-living species *P. cosmopolitanus* and *P. asymbioticus*, which are ubiquitous and frequently abundant members of freshwater bacterioplankton [41,42,43] in habitats that vary in chemical and climatic conditions [44,45,46]. As 16S rRNA data does not provide sufficient resolution for the identification of *Polynucleobacter* species [47], the composition and ecological function of these highly abundant organisms in the marshes of the Everglades warrants further detailed studies. Cultured strains of *P. asymbioticus* originate mainly from humic-rich habitats, where they can utilize products of photodegradation of humic substances [42,48]. The high relative abundance of *Polynucleobacter* in the Everglades watershed is likely explained by the prevailing high concentrations of humic substances, which compose about 50% of the DOC in this environment and are in part mineralized by solar radiation [49].

*Verrucomicrobia*, which were mostly represented by the order *Chtioniobacterales* in the class *Spartobacteria*, had a relative abundance of 7.3% in the ENP freshwater marshes (Figure 3). These bacteria are present in a large variety of terrestrial and aquatic ecosystems and are also a dominant group in many humic lakes, composing up to 19% of the respective BCCs [50,51,52]. Most *Verrucomicrobia* are specialized in the degradation of algal polymers, specifically polysaccharides, such as cellulose and chitin [53]. Therefore, their occurrence is usually correlated with the biomass dynamics of phytoplankton, including *Chrysophyceae* [54], a group that was also present in the freshwater stations in ENP. There were many similarities between the BCC of the ENP transects and that of the Brazos and Mississippi Rivers that also flow into GoM, including high relative abundances of *Limnohabitans*, *Polynucleobacter*, acI clade, LD28 and others [19,55,56]. 

*Cryptophyta* have mostly been considered autotrophic, but exceptions were reported for cryptophytes in ice-covered lakes in Antarctica, where mixotrophic behavior for survival under light-limited conditions was observed during winter [57,58]. Williams and Trexler [59] demonstrated the importance of ‘detrital’ carbon flow in the Everglades, indicating that the microbial loop provides a major route of energy flow to higher trophic levels. The most abundant group of cryptophytes in the marshes, *Chilomonas*, are heterotrophs that can feed on detritus in the form of particulate organic matter [60] and could therefore be a key taxon in the carbon cycle of the Everglades.

Fungi are considered important heterotrophic degraders in periphytic communities [15,61] and could contribute to same processes in the water column. The highest relative abundances of Fungi were observed in the freshwater marshes (8.3% in SRS1d and 10.6% at TS/Ph-2). The most abundant fungal OTU (OTU29) could only be classified as a member of the division *Glomeromycota*, which includes all species involved in arbuscular mycorrhizal symbioses. We cannot exclude the possibility that these sequences come from spores rather than metabolically active cells.

### 4.2. Ecotone

Ecotones are transitions between ecosystems that are characterized by steep environmental gradients [62]. They are critical in regulating the transport of DOM and nutrients into coastal waters [63,64,65]. The hydrographic changes in the Everglades over the last 100 years have had the most impact on its ecotones [66]. 

Mangrove plant roots excrete organic compounds and release oxygen to the rhizosphere, thus changing the chemical characteristics in the sediment area around the roots [67,68,69]. Nevertheless, mangrove sediments are primarily anaerobic with a thin aerobic sediment layer on top [69,70], thus providing chemical characteristics which encourage anoxygenic photolithotrophic and chemolithotrophic sulfur-oxidizing bacteria, which in this study were also found in the water column of the SRS ecotone (e.g., *Ectothiorhodospirales, Thiotrichaceae* and SUP05 cluster). Our results indicate that anaerobic processes in the rhizosphere have a high impact on microbial communities in overlaying water column.

Diatoms showed increased diversity and relative abundance towards the ecotone in our dataset, most notably along the SRS transect (Figure 5). Concomitant with the increase of relative abundance in diatoms, an increase in the abundances of bacterial taxa that have been shown to be associated with diatom blooms, like *Flavobacteriales* and *Rhodobacterales* [71] was also observed (Figure 4). Representatives of *Flavobacteriales* are well-known degraders and consumers of high-molecular-mass organic matter [53,72,73], and *Rhodobacterales* are known to utilize exopolymer particles [74] and could therefore play an important role in the degradation of these compounds in the ecotones in the Everglades.

### 4.3. Florida Bay

FB is a large and shallow estuary, with an average depth of only 1.5 m [75]. In our dataset, the samples from the FB are represented by stations TS/Ph-9, TS/Ph-10 and TS/Ph-11 (Figure 1). The bay receives freshwater runoff from the Everglades marsh mainly through the C-111 Canal and Taylor Slough. Its west side opens to the GoM, the main source of phosphorous for FB, resulting in a phosphorous gradient between the eastern and western parts of FB [76]. Drainage canals are a major source of contamination to the local reef environments, and the nutrients that leak into the estuaries in South Florida lead to occasional/regular algal blooms. Florida Bay waters have been divided into six segments based on their biogeochemical characteristics [77,78]. Our data are in line with some of these general features, as the northern part of the bay (TS/Ph-9) has higher nutrient concentrations, and the central part of the bay (TS/Ph-10) has higher Chl *a* level (Table 1). The highest bacterial cell counts and BP within the FB were also observed in TS/Ph-10 (2.8 × 10^6^ cells L^−1^; 13.8 μg-C L^−1^ d^−1^), surpassing the corresponding values of TS/Ph-9 (1.5 × 10^6^ cells L^−1^; 4.0 μg-C L^−1^ d^−1^).

The highly variable abundance of certain taxa within the FB sites indicates that a higher spatiotemporal resolution is needed to accurately describe the composition of microbial plankton communities in these habitats. The northeastern part of the bay (TS/Ph-9) had higher inorganic nutrient concentrations (Table 1) and autotrophic diatoms dominated the EMC; *Cyclotella* (OTU4, 27.4%) and *Chaetoceros* (OTU11, 17.8%) were the most abundant genus-level taxa. The accompanying BCC contained a large fraction of SAR11 clade (22.2%) and *Rhodospirillales* (15.5%) that specialize in the active uptake of low-molecular weight monomers during diatom dominated phytoplankton blooms [71,79]. *Flavobacteriales* made up 9% of BCC at TS/Ph-9 and might be key players in the initial degradation of organic matter derived from the observed algae [53,72,73]. The closest station to the Gulf of Mexico, TS/Ph-11, exhibited a high abundance of the NS5 marine group (*Flavobacteriaceae*, 23.7%), *Roseobacter* (12.2%) and *Oceanibaculum* (SAR116 clade, 8.3%), which are also abundant in northern parts of the Gulf of Mexico [19,80].

## 5. Conclusions

Pelagic microbial communities differed significantly in habitats along the salinity gradient in the Florida Coastal Everglades. In the freshwater marshes, detrital ‘brown’ carbon flow is essential for the food web and, accordingly, the most abundant organisms were heterotrophs that are presumably capable of degradation of complex organic carbon. In these habitats, solar radiation generates dissolved organic matter via photo-dissolution of flocculent, detrital material and terrestrial humic-like components that can contribute up to 70% of the chromophoric DOM [81]. *Polynucleobacter*, the most abundant prokaryotic group that we detected in the marsh samples, have been shown to utilize products of photodegradation humic substances—a capability that might explain their abundance in this ecosystem. Potentially phagotrophic cryptomonads of the genus *Chilomonas* were identified as predominant eukaryotic microorganisms, indicating that they could hold a key position in this ecosystem by transferring detritus-based biomass to higher trophic levels. In wetland ecotones, oxygen production and excretion of organic matter by mangrove roots, as well as the increased concentration of sulfate from marine waters, create niches for aerobic as well as anaerobic microbial communities, even in the water column. We identified photolithotrophic and chemolithotrophic sulphur-oxidizing bacteria (*Ectothiorhodospirales* and *Thiotrichales*) as predominant members of BCC at the SRS ecotone. Similarly, marine microbial communities at the three sampling sites in Florida Bay were heterogeneous, indicating the presence of (micro)niches and the need of higher resolution of sampling sites. Further data is necessary to pinpoint these trends and to analyze the potential seasonality of microbial communities in these systems. Nevertheless, the datasets presented here will provide valuable baseline information for environmental monitoring in these habitats.

## Figures and Tables

**Figure 1 microorganisms-10-00215-f001:**
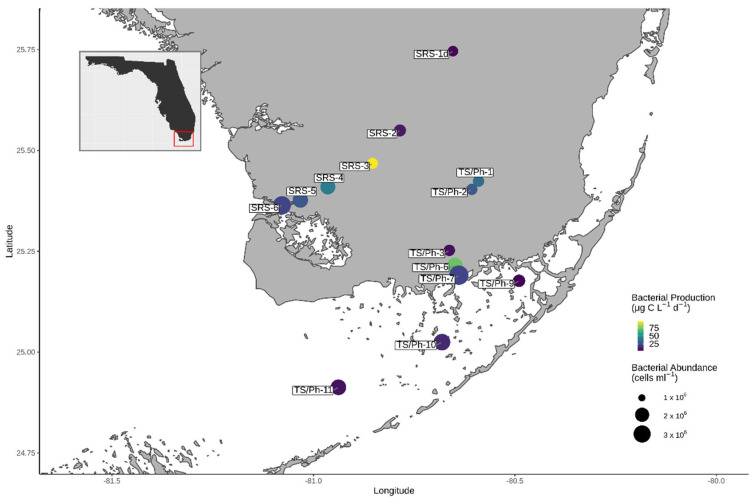
Map of the study area and sampling sites in Everglades National Park, Florida (USA), covering Shark River Slough (SRS 1d-5), Taylor Slough (TS/Ph 1–7) and Florida Bay (TS/Ph 9–11). The size and color of site markers indicate bacterial abundances and bacterial production, respectively.

**Figure 2 microorganisms-10-00215-f002:**
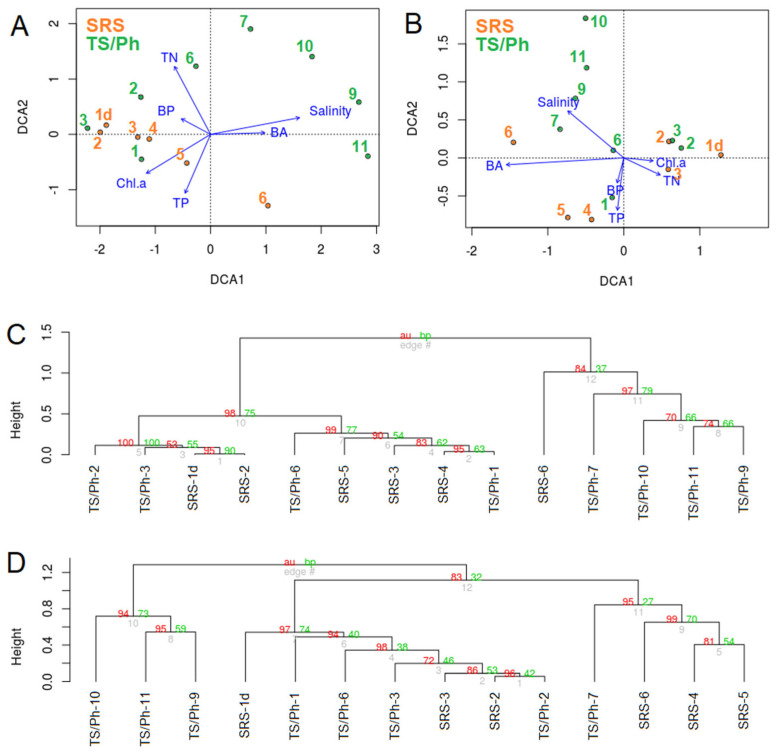
Detrended correspondence analyses of the prokaryotic (**A**) and eukaryotic (**B**) communities fitted with environmental vectors: bacterial abundance (BA), bacterial production (BP), total nitrogen (TN), total phosphorous (TP) and chlorophyll *a* (Chl a) (Table 1). Cluster dendrograms of the prokaryotic (**C**) and eukaryotic (**D**) communities based on sequence variant level composition (cluster method: average; distance: correlation; 1000 bootstrap replications), supplemented with approximately unbiased *p*-values (au, red), bootstrap probability values (bp, green), and edge numbers (grey).

**Figure 3 microorganisms-10-00215-f003:**
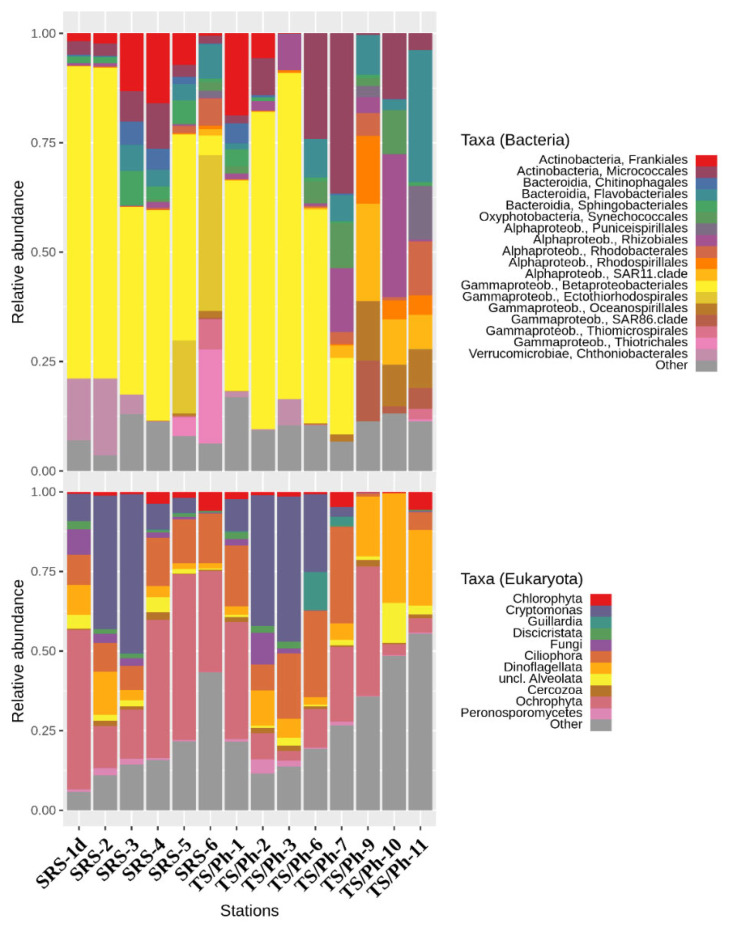
Bacterioplankton community composition on order-level taxa (**top**) and eukaryotic microbial community composition represented by variable higher taxa (**bottom**). In both cases, only taxa that contributed to more than 1% of the respective sequence datasets were included.

**Figure 4 microorganisms-10-00215-f004:**
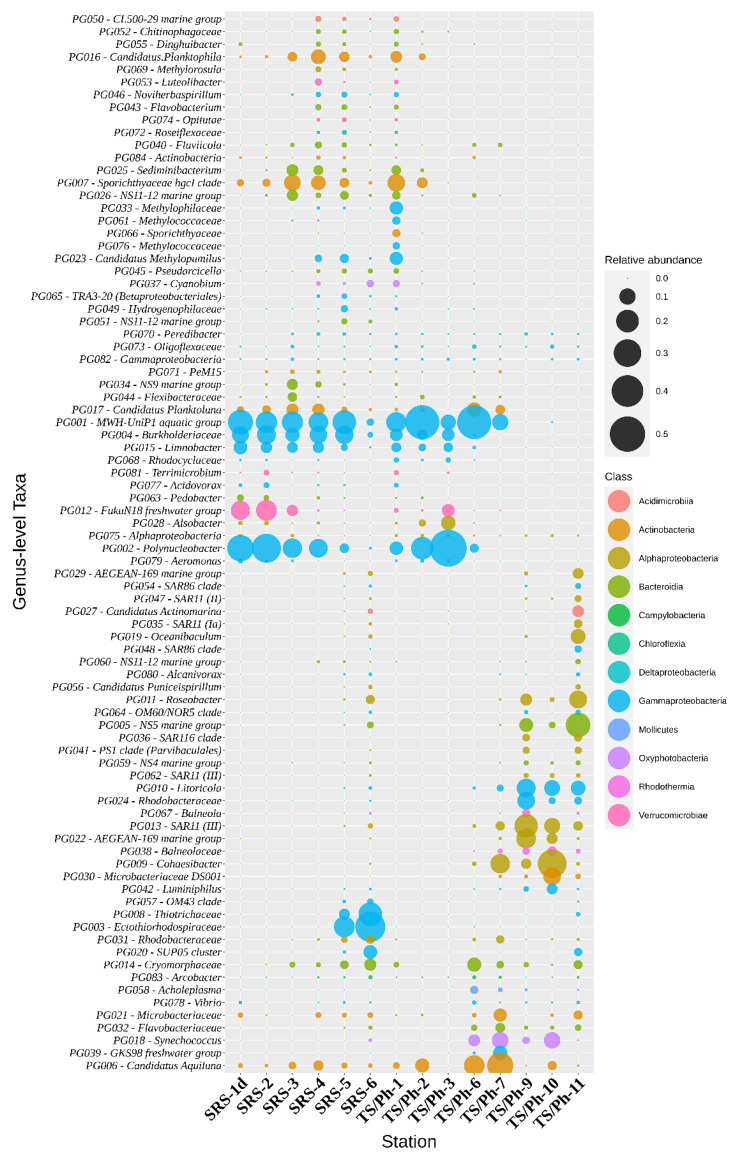
Relative abundance of the bacterial tax comprising more than 0.1% of the bacterial sequences in each sample. Taxa are classified to genera (prokaryotic genera, PG, QIIME2 classifier level 6) or the lowest rank possible according to the classifier. The size of the bubbles represents their relative abundance, and the color indicates the class.

**Figure 5 microorganisms-10-00215-f005:**
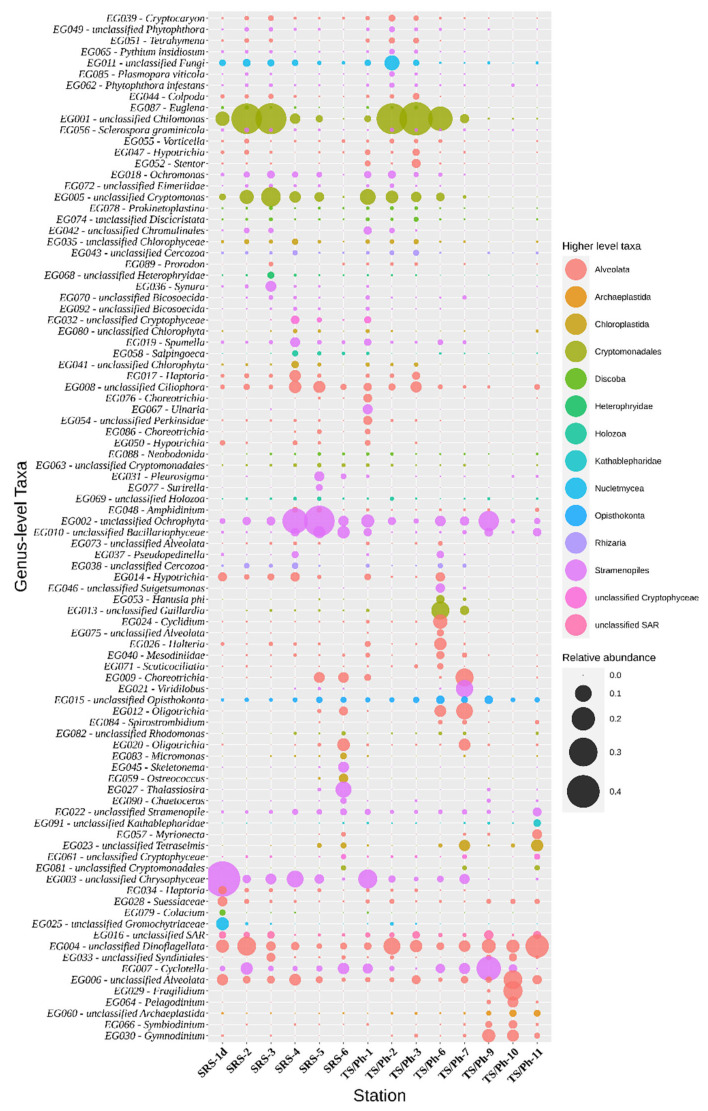
Relative abundance of eukaryotic taxa comprising more than 0.1% of the bacterial sequences in each sample. Operational taxonomic units (OTU) are classified to the lowest rank possible according to the classifier. The size of the bubbles represents their relative abundance, and the color indicates the higher-level classification, as depicted on the right. Percentages in parentheses are sequence similarity values to the listed taxon.

**Table 1 microorganisms-10-00215-t001:** Station locations, bacterial abundances (BA) and bacterial productivities (BP) and selected environmental data (Chlorophyll a—Chl a; total nitrogen—TN; total phosphorous—TP).

Station	BA (cells mL^−1^)	BP (μg C L^−1^ d^−1^)	Chl a (μg L^−1^)	Salinity (PSU)	TN (μM)	TP (μM)	DOC (μM)
TS/Ph-1	1,206,442	36.53	2.89	0.2	33.67	0.6	851.67
TS/Ph-2	1,200,550	29.99	0.11	0.2 *	43.70 *	NA	746.67
TS/Ph-3	1,233,549	6.54	2.71	0.2	66.23	0.53	893.33
TS/Ph-6	2,586,481	71.87	1.16	6.1 *	75.40 *	0.4 *	1254.17
TS/Ph-7	3,884,709	22.58	1.56	14.8 *	68.84 *	0.41*	1085.83
TS/Ph-9	1,493,439	3.98	0.07	39.50	45.39	0.23	460.42 **
TS/Ph-10	2,811,578	13.82	0.12	41.40	53.62	0.23	631.67 **
TS/Ph-11	2,455,006	7.58	0.06	39.49	23.25	0.30	270 **
SRS-1d	1,048,253	5.11	2.49	0.2	70.29	0.54	1268.33
SRS-2	1,390,989	9.51	1.63	0.1	55.93	0.37	1148.08
SRS3	1,127,995	95.80	1.25	0.2	53.25	1.00	1243.33
SRS4	2,272,470	42.85	1.79	0.4	63.76	0.71	1157.33
SRS5	2,381,611	29.66	1.61	2.1	51.76	0.63	1066.67
SRS6	3,272,441	22.64	1.3	19.3	37.94	0.70	929.25

* Measurement taken 1–3 days apart from collection of water sample for microbial analyses. ** TOC (μM), DOC data not available. NA = not analyzed.

**Table 2 microorganisms-10-00215-t002:** Total number of analyzed sequences, observed sequence variants (SV) and genera-level taxa (Qiime2 classifier, level 6), as well as Shannon diversity indices (H’) for 16S rRNA (bac.) and 18S rRNA (euk.) gene data.

Station	Sequences (bac.)	SV (bac.)	Genus-Level Taxa (bac.)	H’ (bac.)	Sequences (euk.)	SV (euk.)	Genus-Level Taxa (euk.)	H’ (euk.)
TS/Ph-1	80125	281	151	6.36	173981	1654	213	8.41
TS/Ph-2	70198	1302	165	4.58	269629	1302	148	7.02
TS/Ph-3	90170	1227	145	4.19	198954	1227	135	7.02
TS/Ph-6	68729	752	87	4.75	305749	752	115	6.02
TS/Ph-7	50720	1008	75	4.61	208940	1008	154	6.71
TS/Ph-9	56338	488	50	4.51	77764	488	103	4.83
TS/Ph-10	53636	1084	64	3.85	102902	281	52	5.50
TS/Ph-11	170185	1654	128	5.82	135226	1084	146	6.81
SRS-1d	56789	1174	104	4.66	377682	1174	130	5.34
SRS-2	58820	1029	86	4.42	183907	1029	141	6.42
SRS-3	76305	1358	168	5.81	273752	1358	172	6.20
SRS-4	97050	490	139	5.88	389092	490	95	6.65
SRS-5	149420	1748	188	5.71	315255	1748	203	6.60
SRS-6	129150	1252	133	4.44	188495	1252	188	6.77

## Data Availability

Demultiplexed raw data was submitted to the Sequence Read Archive under accession number PRJNA525456.

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
