# Peer review of "Water Column Microbial Communities Vary along Salinity Gradients in the Florida Coastal Everglades Wetlands"

_microorganisms, 2022, doi:10.3390/microorganisms10020215_

Round 1

Reviewer 1 Report

  1. It would be helpful if the authors include somewhat more Discussion about the implications for the sampling time of year (August) during the tropical wet season, versus other times. Would there likely be differences in their findings at other times of the year, and what value if any would there be for further research at these alternate times of the year?
  2. Can the authors explain more explicitly why the TS/Ph samples extended into the Florida Bay, but the SRS samples largely ended short of the margin of the everglades and the edge of the Gulf of Mexico? I presume it is clearly due to the geomorphology and ecotones involved, etc. given the margins of the peninsula near the Gulf are likely much less shallow, etc. But, it would be helpful if a brief statement was added in the methods. The Discussion provides more information on the geomorphological aspects of the terminus of the TS/Ph in the FB, but it would be helpful if more comments on the geomorphological and environmental attributes of the  terminus of the SRS near the Gulf of Mexico were briefly explained. Is the terminus of SRS at or near the Ponce de León Bay?
  3. In this regard, it might be helpful if the authors give approximate geographic coordinates for the first sampling sites (SRS-1 & TS/ph-1) and the end sampling site on the peninsula at the SRS-6 and TS/ph-7 simply to allow the reader to be certain where these important geographic locales are.
  4. Lines 219 -224, the data suggest that the ecosystems may be largely heterotrophic, especially near the terminus of each freshwater system, given the bacterial loads, and large differences in the mixo- or heterotrophic eukaryotes, compared to the Chlorophyta. However, without more information it is difficult to judge. Some comment on this would be helpful. Some estuaries, for example, are typically heterotrophic due to heavy flocculent matter and low light penetration, etc. The decreasing Chl a data and increasing bacterial abundance suggests increasingly heterotrophic conditions closer to the end points of the two rivers.
  5. Lines 281-282 listing relative increases in Ochrophyta (some possibly mixotrophic) and Ciliophora (largely heterotrophic) would further suggest the two systems become increasingly more heterotrophic toward their endpoints, which would not be unusual. As the authors acknowledge, some dinoflagellates are also ,mixotrophic and prey on other microeukaryotes, and the cryptophyte, or more particularly a cryptomonad, (Chilomonas) as the authors acknowledge is heterotrophic.
  6. The taxon ‘Ochrophyta’ is not so well defined in some research treatments (or encompasses a rather broad group of microeukaryotes), and it would be helpful if the authors indicated what were some of the major genera in their designation for this taxonomic category. This would help to get a better grasp of the ecological role for this group as used in this research.
  7. The Shannon diversity indices (H’) levels (Table 2) are noticeably high throughout the stations as might be expected in productive systems of this kind, but it would be interesting if the authors commented on this briefly in the Discussion.
  8. Line 335 Burkholderiales, not Burholderiales

Overall, this is interesting and provides important base line data for this potentially seriously affected ecosystem if continued climate change effects are not mitigated.

Author Response

  1. It would be helpful if the authors include somewhat more Discussion about the implications for the sampling time of year (August) during the tropical wet season, versus other times. Would there likely be differences in their findings at other times of the year, and what value if any would there be for further research at these alternate times of the year?

We are currently working on a long-term dataset and will address this in more detail in a separate paper. We added a statement to the conclusions at lines 448-450 stating that further data is needed to analyze potential seasonality of microbial communities in these systems.

  1. Can the authors explain more explicitly why the TS/Ph samples extended into the Florida Bay, but the SRS samples largely ended short of the margin of the everglades and the edge of the Gulf of Mexico? I presume it is clearly due to the geomorphology and ecotones involved, etc. given the margins of the peninsula near the Gulf are likely much less shallow, etc. But, it would be helpful if a brief statement was added in the methods. The Discussion provides more information on the geomorphological aspects of the terminus of the TS/Ph in the FB, but it would be helpful if more comments on the geomorphological and environmental attributes of the terminus of the SRS near the Gulf of Mexico were briefly explained. Is the terminus of SRS at or near the Ponce de León Bay?

For this study, we piggy-backed on the routine sampling of the FCE LTER teams. The sites are the FCE LTER core sampling sites. The project has been going on since 2000 and we do not know about the history of the site selection.  We now added a link to the methods on line 97-98 that leads the readers to a detailed description of the sites.

  1. In this regard, it might be helpful if the authors give approximate geographic coordinates for the first sampling sites (SRS-1 & TS/ph-1) and the end sampling site on the peninsula at the SRS-6 and TS/ph-7 simply to allow the reader to be certain where these important geographic locales are.

Coordinates of the sites can be found at https://fce-lter.fiu.edu/research/sites/index.php

This link was added to the methods on lines 97-98.

  1. Lines 219 -224, the data suggest that the ecosystems may be largely heterotrophic, especially near the terminus of each freshwater system, given the bacterial loads, and large differences in the mixo- or heterotrophic eukaryotes, compared to the Chlorophyta. However, without more information it is difficult to judge. Some comment on this would be helpful. Some estuaries, for example, are typically heterotrophic due to heavy flocculent matter and low light penetration, etc. The decreasing Chl a data and increasing bacterial abundance suggests increasingly heterotrophic conditions closer to the end points of the two rivers.

A previous version of this manuscript was rejected after review by a different journal because of speculations on potential metabolism of the microbial communities purely based on 16S data. Therefore, we do not want to include a statement on this without further data.  

  1. Lines 281-282 listing relative increases in Ochrophyta (some possibly mixotrophic) and Ciliophora (largely heterotrophic) would further suggest the two systems become increasingly more heterotrophic toward their endpoints, which would not be unusual. As the authors acknowledge, some dinoflagellates are also ,mixotrophic and prey on other microeukaryotes, and the cryptophyte, or more particularly a cryptomonad, (Chilomonas) as the authors acknowledge is heterotrophic.
  2. The taxon ‘Ochrophyta’ is not so well defined in some research treatments (or encompasses a rather broad group of microeukaryotes), and it would be helpful if the authors indicated what were some of the major genera in their designation for this taxonomic category. This would help to get a better grasp of the ecological role for this group as used in this research.

Unfortunately, in many cases this was the best our classifier could do, and we cannot provide further information. Clearly, there is a large gap in missing molecular information for these taxa.

  1. The Shannon diversity indices (H’) levels (Table 2) are noticeably high throughout the stations as might be expected in productive systems of this kind, but it would be interesting if the authors commented on this briefly in the Discussion.

As this is a snapshot of the microbial communities and is based on an n=1 for each site, we want to keep this discussion for a future paper that contains more data that might corroborate these trends. The diversity might be biased by weather or other conditions and long-term data is really needed to compare the diversity of these communities to those in other habitats.

  1. Line 335 Burkholderiales, not Burholderiales

Changed, thank you.

Reviewer 2 Report

In general this manuscript describes a microbial community investigation in the waters of the Florida Everglades which is timely.  The writing style is good and the results/conclusions are appropriate.  There is some concern about the methods utilized for the ancillary environmental data since the website for the protocols was not responsive (fttp://fcelter.fiu.edy/data/core/index.htm) effectively making it unavailable.  This is a serious problem for any researcher to know the methods utilized and to reproduce the research.  If this problem is eliminated then the conclusions seem reasonable especially if additional higher resolution sampling in Florida Bay occurs in followup studies.     

Author Response

We want to thank the reviewer for the positive comments and apologize for the outdated link. We now corrected the link to:

https://fcelter.fiu.edu/data/protocols/index.html

on line 95 in the revised manuscript.